# Single-Molecule Counting of Nucleotide by Electrophoresis with Nanochannel-Integrated Nano-Gap Devices

**DOI:** 10.3390/mi11110982

**Published:** 2020-10-31

**Authors:** Takahito Ohshiro, Yuki Komoto, Masateru Taniguchi

**Affiliations:** Institute of Science and Industrial Research, Osaka University, 8-1 Mihogaoka, Ibaraki, Osaka 567-0047, Japan; toshiro@sanken.osaka-u.ac.jp (T.O.); komoto@sanken.osaka-u.ac.jp (Y.K.)

**Keywords:** nanochannel, DNA, single-molecule detection

## Abstract

We utilized electrophoresis to control the fluidity of sample biomolecules in sample aqueous solutions inside the nanochannel for single-molecule detection by using a nanochannel-integrated nanogap electrode, which is composed of a nano-gap sensing electrode, nanochannel, and tapered focusing channel. In order to suppress electro-osmotic flow and thermal convection inside this nanochannel, we optimized the reduction ratios of the tapered focusing channel, and the ratio of inlet 10 μm to outlet 0.5 μm was found to be high performance of electrophoresis with lower concentration of 0.05 × TBE (Tris/Borate/EDTA) buffer containing a surfactant of 0.1 *w/v*% polyvinylpyrrolidone (PVP). Under the optimized conditions, single-molecule electrical measurement of deoxyguanosine monophosphate (dGMP) was performed and it was found that the throughput was significantly improved by nearly an order of magnitude compared to that without electrophoresis. In addition, it was also found that the long-duration signals that could interfere with discrimination were significantly reduced. This is because the strong electrophoresis flow inside the nanochannels prevents the molecules’ adsorption near the electrodes. This single-molecule electrical measurement with nanochannel-integrated nano-gap electrodes by electrophoresis significantly improved the throughput of signal detection and identification accuracy.

## 1. Introduction

The nanochannel is a key part of micro/nano-machine devices that is widely used in various research fields, such as the analytical sensing and/or mechanical engineering fields, because it provides indispensable functions such as sample-introduction, separation, and purification [1,2]. One of the most attractive research fields is micro total analytical systems (µTAS), in which the chemical analysis system is miniaturized by micromachining, and nanochannels seamlessly combine various kinds of functional units with various sensors so that they provide the developments of medical application fields such as blood analysis and drug evaluation systems [3,4,5]. 

Single-molecule measurement became an important analytical tool for understanding biological phenomena. Among them, a single-molecule electrical measurement method by using nanostructures such as nanopores and nanogap electrodes is attracting attention because of its advantages such as simplicity and integration by miniaturization [6,7,8]. We have proposed single-molecule measurement using nano-gap electrodes for sequencing and biomolecule identification [9,10,11,12]. The detection principle is as follows: When sample molecules pass between the nano-gap electrodes that work as sensors, a tunnel phenomenon induces electron transfer through the molecules, resulting in the detection of individual electrical conductivity due to the electronic state of each of the molecules. Furthermore, the signal time-profiles during passing through the nano-gap represents the characteristic molecular behavior due to the interaction between the molecule and the electrode, which also serve to identify various kinds of the molecule [13,14,15,16]. As a result, we have succeeded in measuring various biomolecules such as DNA, amino acids, and neurotransmitters [9,11,16]. In this principle, the driving force of sample introduction through the sensor nanogap electrode is the diffusion into the sensor area due to Brownian motion, resulting in the sparse frequency of the signals. Therefore, throughput improvements of the method by using nano-gap devices are one of the next important steps.

In this study, we developed a nanochannel-integrated nano-gap device that can control the fluidity of biomolecules by electrophoresis. Since in the nanospace below 100 nm, like this nano-gap structure, hydrostatic pressure of solvent flow by a pump do not serve an active introduction of sample molecules [17,18], this active introduction by electrophoresis can improve throughput of sample molecules and stabilize the detection frequency. There are a few issues for electrophoretic introduction of sample molecules by nanochannel-integrated nano-gap devices. Among them, it is expected that electro-osmotic flow (EOF) in the nanochannel can hinder the introduction of charged samples around the inlet area between chamber and nano-channel [1,19,20]. In order to reduce or control it, we addressed it by the following approaches. 

First, we reduced the EOF in the nanochannel by modification of nanochannel surfaces because EOF is the motion of liquid induced by net charge movements in the electrical double layer on the surface of the channels upon application of an electric field around the channel of inlets. We added a small amount of surfactants into the sample solution, and found a significant improvement of sample introduction by reduction of EOF around the inlet of the nanochannel. Second, we improved the electrophoretic force around the inlet of the nanochannel by its optimization of “tapered focusing channel” parts that were connecting parts between the nanochannel and sample chamber. 

By using this nanochannel-integrated nanogap device, we performed single-molecule measurement for a DNA base, i.e., deoxyguanosine 5′-monophosphate (dGMP), under the electrophoresis condition, and succeeded in detection of around a few hundred signal molecules per second. This improved the measurement throughput of this method by at least more than ten times compared to that without electrophoresis and succeeded in stabilizing the signal detection frequency. In addition, it was also found that the long duration time signals, which are difficult to identify in signal analysis, were significantly reduced. This is due to the reduction of non-specific adsorption to the sensor electrode by improvement of sample fluidity, including electrophoresis and EOF effect around the nanochannel. From this, the single-molecule electrical detection method by electrophoresis, which combines nanogap and nanochannel, is an effective method that enables stable high-throughput measurement of single-molecule measurement, and it would be applicable to a wider range of biomolecule identification measurement and understanding these behaviors and phenomena at the single-molecule level.

## 2. Materials and Methods 

### 2.1. Fabrication of Nanochannel-Integrated Mechanically Controllable Break Junction (MCBJ) Devices for Single-Molecule Detection

The nanogap electrodes were constructed from nanofabricated MCBJs. The procedures for fabricating the MCBJs are detailed elsewhere [21,22]. In this study, we integrated a nanochannel into the nanogap device (Figure 1a) as follows. The device fabrication is shown in below. We designed the nanochannel with 200 nm width and 8 μm length and tapered focusing channel with 2 μm inlet/0.2 μm outlet and 5 μm length (Figure 1b). A cover made of polydimethylsiloxane (PDMS) is attached to the silicon substrate. The PDMS cover has a microchannel that connects the hole for introducing the sample solution and the nanochannel of the sensor in advance (Figure 1c). PDMS are purchased from Torei Dow Corning. Finally, the PDMS cover and silicon substrate are treated with ozone plasma and then bonded. The electrodes used in electrophoresis (Figure 1d) are prepared by electrochemical oxidation of silver wires as follows: A silver wire (The Nilaco Corporation, Tokyo, Japan) is electrochemically oxidized in 1 M NaCl by using the Electrochemical Analyzer Model 1030 (ALS Co., Ltd., Tokyo, Japan). The resistance of the prepared Ag/AgCl electrode was found to be around 20 kΩ. The formed gold nano-junction is broken by the MCBJ systems and the distance is set to 0.6–0.8 nm by the piezo element. During the measurement, a gap-distance was kept by feedback control of the piezo actuators. The device fabrication is shown in Figure 1e.

### 2.2. Fabrication of Nanochannel Device for Single-Sample Observation by Optical Imaging

For optical observation by optical microscope, a silicon wafer was used for a silicon device substrate, and a glass wafer (SW-YY, ~300 mm thick, Asahi Glass Co., Tokyo, Japan) was used for its cover substrate. The device fabrication is shown in Figure 1d. First, a pair of through-holes for the inlet/outlet of sample solution is created on the silicon substrate. A mask Cr layer is formed on a silicon substrate by radio frequency (RF) sputtering. After that, the photoresist is spin-coated, and the photoresist is drawn. After that, the Cr film is removed by a wet-etching, and a through-hole is formed by the Bosch method for the flow path hole for the inlet/outlet. In the next step, a microchannel is fabricated on the substrate. After Cr layer formation, the photoresist is spin-coated, and the microchannel pattern is formed by photolithography. Then, the formed pattern is etched by reactive ion etching (RIE), and the Cr layer is removed by wet-etching. In the next step, a nanochannel is fabricated. An electron beam resist is spin-coated, and the developed nanochannel pattern is formed by electron beam lithography, and then reactive ion etching (RIE) is performed to form nanochannels. Finally, a cover glass is fused with the nanochannel substrate by the anodic bonding method as follows [23]: Both the glass wafer and the microfabricated Si substrate were cleaned in a piranha solution (H_2_SO_4_:H_2_O_2_ = 3:1) for >1 h at 100 °C, rinsed by Milli-Q water, and dried by N_2_ gas blow. Then, the pre-cleaned wafers were sandwiched between a pair of carbon-plated electrodes. The assembly was heated at 300 °C on a hotplate, followed by applying a dc voltage of >+400 V on the electrode in contact with the Si wafer.

### 2.3. Imaging of Fluorescence-Stained λDNA Molecules and Fluorescence Particle Image by Fluorescence Optical Microscope

The imaging was performed using inverted fluorescence optical microscopes (AM TIRF MC Leica Microsystems) equipped with a 100 times oil immersion objective lens (Leica Camera AG, Wetzlar, Germany) (Figure 2a–c). The images were acquired by an iXon3 897 electron-multiplying Charge-coupled device (EM-CCD) Camera (Andor Technology Ltd., Belfast, Northern Ireland). Each of the images was obtained at the rate of 10.6 frames per second. A 488 nm diode laser was used to excite the fluorescent objectives. For λDNA imaging, the sample λDNA was purchased (NIPPON GENE CO., LTD., Tokyo, Japan) and used without further purification. The λDNA was stained with YOYO-1™ Iodide (Thermo Fisher Scientific, Waltham, MA, USA) and dissolved into 0.1 × Tris-Borate- ethylenediaminetetraacetic acid (TBE) or 0.05 × TBE containing 0.1 *w/v*% polyvinyl-pyrrolidone (PVP). We used TBE with pH 7.8 as a buffer for the solution of optical imaging of particles and λDNA, and single-molecule measurements. The concentration of λDNA was 0.1 μg/mL. The TBE buffer solutions are effective for slightly basic conditions, which keep DNA deprotonated and soluble in water. The sample λDNA solution and the buffer solution were introduced into one and the other into two chambers in the nanochannel device respectively, and then voltage was applied between chambers. The flow behavior of the sample λDNA inside the channel of the devices was observed under the fluorescence microscope. Basically, the sample solutions are inserted in the channel region of the device by the capillary force. In order to assist the force, the surface of the nanochannel was treated with ozone plasma cleaning. Under the no electrophoresis condition, we found that the movement of particles are mainly induced by head pressure inside the channel and, around nanochannel-integrated nano-gap electround, the movement DNA molecules would be due to Brownian motion. YOYO-1 iodide (491 nm absorption, 509 nm emission) as the DNA backbone staining dye was excited by 488 nm laser diode, and the fluorescence was obtained through optical filters. For single-particle imaging, 40 nm polystyrene florescent particles (FluoSpheres™ Carboxylate-Modified Microspheres yellow-green fluorescent, ThermoFisher Co. Ltd., Waltham, MA, USA) were purchased, and used for a sample optical imaging without further purification. The sample solution of polystyrene florescent particles was diluted in 0.1 × TBE solution containing 0.1 *w/v*% polyvinyl-pyrrolidone (PVP) and the concentration was set to around 2 × 10^12^ particles/mL. The florescent particle (505 nm absorption, 515 nm emission) was excited by a 488 nm laser diode. The fluorescence images were analyzed by Image J software version 1.52.

### 2.4. Procedure for Current Measurements 

Tunnel current measurements were performed in 1 µM deionized aqueous solutions of sample deoxyguanosine monophosphate (dGMP) (Sigma-Aldrich, St. Louis, MO, USA) 0.05 × TBE containing 0.1 *w/v*% polyvinyl-pyrrolidone (PVP). Sample solutions were dropped (20 µL for each measurement) on the center of the sensor plate where gap electrodes were fabricated. Gap size was set to 0.7 nm and finely tuned by the piezoelectric element during all the measurements. The current across the electrodes was amplified by a custom-built logarithmic current amplifier, and recorded at 10 and 100 kHz using a NI PXIe-4081 digital multimeter (National Instruments, Austin, TX, USA) and NI PXI-5922 oscilloscope (National Instruments, Austin, TX, USA) under a direct current (DC) bias voltage of 0.1 or 0.4 V. After every 1 h of current-time (i-t) measurement, we replaced the MCBJ sample with a new one. The measurements were carried out in more than three sets using different gold gap sensors.

## 3. Results

### 3.1. Direct Observation of Sample Behaviors in Nanochannel Device

We observed single-molecule behaviors in a λDNA aqueous solution containing a 0.05 × TBE buffer by optical microscope under electrophoresis conditions (Figure 2a). From the results under the application of electrophoresis, it was found that the behavior of sample λDNA was unstable inside tapered focusing nanochannel parts and backflow occurs especially around the inlet of the nanochannel, resulting in prevention of introduction of sample λDNA into nanochannel regions. It seems that the backflow is a typical EOF that is induced by the movement of the solvent by the counter ions of the nanochannel surface charge. Therefore, in order to suppress the negative charge of SiO_2_ on the surface, we utilized a sample solution containing 0.1 *w/v*% polyvinyl-pyrrolidone (PVP). Although several kinds of surfactants, such as polyethylene-glycol, polyvinyl-pyrrolidone, polyvinyl-alcohol, and Triton X-100, are reported in order to suppress the negative charge inside the nanochannel [24,25], surfactants with a π-conjugated ring are not suitable for this study because the size of surfactants with a π-conjugated ring is comparable to those of DNA and RNA bases, which have a pyrimidine or purine ring, so that signals of surfactant molecules itself could resemble those signals of DNA and RNA samples. In addition, since an addition of a large amount of surfactant could induce a dissociation or denaturation of analyte, the concentrations of surfactant should be optimized for the imaging of λDNA. Until now, it has been reported that the concentration in the several *w/v*% PVP is used for DNA sample imaging under electrophoresis [26,27] and it was also found that even the small addition, such as 0.1 *w/v*% PVP, significantly reduced the EOF inside the channel, relative to an addition of 0% PVP [28]. Therefore, we investigated a sample solution containing 0.1 *w/v*% PVP for the EOF suppression condition in the following. 

Figure 2b,c show a typical observation for a λDNA aqueous solution containing 0.1 *w/v*% PVP and for that without PVP, respectively. The DNA molecules were successfully introduced into nanochannel regions so that PVP effectively suppressed the generation of EOF inside devices (Figure 2b and Appendix A). On the other hand, EOF are also observed for that without PVP (Figure 2c and Appendix A). In order to track the movement of DNA molecules through the channel [29,30], a typical trace plot is shown in Figure 2d. From the statistical analysis for ten particle traces of the optical microscope image, the speed of the DNA in the tapered focusing region was found to be 0.124 ± 0.53 (μm/second) for the DNA solution without PVP and 24.03 ± 2.11 (μm/second) for the DNA solution containing 0.1 *w/v*% PVP. In general, in the case of the high ion-concentration solution, DNA easily transport through the nanochannel due to the thin electrical double layer. On the other hand, it is reported that single-stranded DNA in the nanochannel in low ionic solutions are elongated inside the nanochannel [31,32,33]. It has been reported that the concentration in the range from 0.5 × TBE to 0.01 × TBE is often used for DNA sample imaging under electrophoresis [34,35]. However, it was also found that the high concentration of the ion induces the electrical noise-level for single-molecule electrical detection so that the lower ion concentration such as 0.05 × TBE buffer and/or 1 mM phosphate buffer is suitable for single-molecule electrical detection [9]. In this study, we used the 0.05 × TBE buffer solutions both for the DNA imaging and single-molecule electrical detection. The stable single-molecule imaging for λDNA samples was successfully performed and the sample smooth translocation behaviors were observed in the TBE buffer solution (Appendix A). In addition, the electrical noise level for the buffer was found to be around 2 pA for root mean square (RMS) of current values so that the TBE concentration is enough for single-molecule electrical detection of the sample dGMP molecules in this study. Therefore, we utilized a sample solution with 0.05 × TBE containing 0.1 *w/v*% PVP for the following measurements.

### 3.2. Dependency of Sample Electrophoretic Behaviors on the Shape of Nanochannel Devices

Next, we observed the translocation behavior of sample molecules/particles from the sample chamber to the other chamber via the nanochannel by optical microscope. When electrophoresis is applied inside a nanochannel device, in which the solution chamber is directly connected to the nanochannel, the aggregation of samples frequently occurred around the inlet of the nanochannel. This would be due to electric field concentration near the nanochannel [36]. Therefore, in order to control the electrical field near the nanochannel intel, we added a “tapered focusing channel” between the nanochannel and sample chamber in the device (Figure 2a) and evaluated the effect of the shape of the tapered focusing channel. As a sample, carboxylated 40 nm fluorescent particles are used for this evaluation. Importantly, the shape of this tapered focusing channel determines and/or influences the local electric field and EOF, resulting in a significant effect on the translocation behavior of samples. We fabricated three types of the nanochannel devices containing a 20 μm length tapered focusing channel, whose reduction ratio of inlet-width (μm) to outlet-width (μm) were 10 μm/5 μm (10/5), 10 μm/5 μm (10/1), and 10 μm/0.5 μm (10/0.5), respectively. By observing the fluorescent particles one by one, a typical time-trace of the bright spot for each of the devices is plotted in Figure 3b. It was found that each of the devices have different speed and acceleration performance, which depend on the degree of the reduction ratio. The location of a particle is defined as x (if x = 0, the location of a particle is the center position of the nanochannel in the device), and the speed of a particle is defined as the first derivative of particle position (dx/dt), and the acceleration of the particle is defined as the second derivative of particle position (d^2^x/d^2^t).

The speed around the nano-gap region (−10 μm < x < 10 μm) was almost constant. From ten trace data of particle x positon, its values were found to be 225 ± 63 μm/s for 10/5, 145 ± 23 μm/s for 10/1, and 45 ± 27 μm/s for 10/0.5, respectively. In general, the velocity, *v,* of a sample particle is determined by the sum of the electrophoresis velocity, *v_ep_*, and the velocity, *v_eo_*, of the electroosmotic flow from the flow path. Since each velocity, i.e., *v_ep_* and *v_eo_*, is a proportional relationship between the DC electric fields *E* and *v*, they are given by the following equation:*v* = *v*_ep_ + *v*_eo_ = (*μ*_ep_ + *μ*_eo_) *E*(1)

It is suggested that a stable electric field is formed inside the nano-gap regions for all the devices, and it was found that the stable flow control is possible.

Next, a typical acceleration (d^2^x/d^2^t) data in the tapered focusing channel region is shown in Figure 3d. In general, the acceleration of a sample is related to the force in the flow inside the nanochannel. From ten trace data of particle x positon, in the acceleration region in the tapered focusing channel (x < −10 μm), each of the maximum acceleration values were found to be 52 ± 30 μm/s^2^ for 10/5, 516 ± 239 μm/s^2^ for 10/1, and 1264 ± 613 μm/s^2^ for 10/0.5, respectively. Importantly, compared to 10/0.1, 10/1 and 10/0.5 have a large acceleration of about two digits so that the force received by channels of 10/1 and 10/0.5 is significantly large. Since the force can serve an entropic dissociation energy of a polymer sample such as DNA upon introduction into the nanochannel, larger acceleration is desirable for measurements of the biopolymer samples. Therefore, we utilized a device containing a tapered focusing channel of 10/0.5 for the following single-molecule electrical measurements.

### 3.3. Evaluation of Molecular Detection and Signal Behaviors of DNA Samples by Electrophoresis

Finally, single-molecule measurement was performed in a 1 μM aqueous solution of dGMP by using the nanochannel-integrated nano-gap electrode (Figure 4a). dGMP molecules are detected current positive signals because the tunnel-currents are induced via the molecules around the nano-gap electrode (Figure 4b). First, in the case of measurements without electrophoresis, the average frequency of the sample molecule detection was found to be 31.5 ± 54.2 (signal count per ten-seconds). This molecular detection driving force is considered to be Brownian motion. Next, we applied electrophoresis between a sample chamber and another chambers as follows: The sensing electrode that is one of the gap electrodes was electrically grounded. Then, direct current (DC) voltage was applied between a grounded sensing electrode (+0 V) and electrophoresis electrodes in the sample chamber and the same DC voltage was simultaneously applied between the grounded sensing electrode and electrophoresis electrodes in the other chamber. When 0.1 V for the sample chamber and −0.1 V for another chamber were applied, the potential difference between the chambers was 0.2 V so that the electrophoresis conditions are described as “0.2 V”. 

Figure 4c,d shows a typical current-time profile for before and after electrophoresis, respectively. When DC voltage was applied inside the nanochannel devices, the detected signal number was increased, and this detection frequency depended on the DC applied voltage. Figure 4e shows the comparison of the detection frequency and DC applied voltage. Importantly, the number of detected molecules increases in proportion to the DC applied bias potential from 0.2 to 1.2 V. Similarly, in the particle detection, the particle velocity is found to increase in proportion to DC applied voltage. Therefore, these results suggest that the flow velocity of molecules is increased by electrophoresis, resulting in increments of the number of detected molecular signals. On the other hand, it was also found that the frequency increment becomes saturated and unstable when the applied voltage becomes 1.6 V or higher. After the application of 1.6 V, it was found that some bubbles appeared at the surface of electrophoresis electrodes. This suggests that the generation of bubbles was due to electrochemical decomposition of water, resulting in the reduction of actual electrode surface or the partial blockage of the nanochannel. From the above results, the application of 1.0 V was found to be the optimal electrophoresis condition for this experimental set-up. In the case of applied voltage of 1.0 V, the frequency of molecule detection increased to around 506 ± 85 (signal count per ten-seconds) so that this throughput of signal detection was significantly improved, relative to that in the case of measurement without electrophoresis. 

This increase in the velocity of the passing molecule by electrophoresis appears in the signal duration-time. Until now, we have reported that current-time profiles of sample signals represent the translocation behavior of the sample molecule near the gap electrode so that the duration-time correlates with the flow rate and interaction of the molecule [11,14]. Therefore, the duration-time length of the signals should be reduced if the translocation speed of the sample is increased. Figure 4f shows the duration-time distribution histogram for electrophoresis (1.0 V), and that for no electrophoresis. The average signal duration time during electrophoresis (1.0 V) of 1.22 milliseconds (ms) was shorter than that of 2.25 ms under no electrophoresis. In the nanochannel or nanopore sensing, clogging of analytes inside the channel/pore could sometimes occur, resulting in inducing the low signal frequency. In this study, there was no dependency of the buffer concentration on signal frequency. Importantly, the proportion of longer duration-time signals (>9 ms) was found to be significantly increased from 9.1% for no electrophoresis, compared to 2.6% for electrophoresis. It is considered that this reflects the decrement of physical adsorption by the interaction between the electrode and the molecule, in addition to increments in the sample speed by electrophoresis. This reduction of the longer duration-time signal detection contributes to accurate discrimination from noise signals because the intensity of longer duration-time signals is sometimes larger than typical intensity of dGMP signals, and the signal behaviors are very similar to signals by the contaminants. 

## 4. Discussion

From these results, single-molecule electrical measurement with a nanochannel device by electrophoresis significantly contributes to the improvement of throughput. We assume that this is due to the control of the balance between EOF and electrophoresis, mainly the suppression of EOF inside nanochannels that transport and introduce sample molecules into sensor regions. Since the acceleration of the particles (Figure 3d) is closely related to the electrophoresis force applied to the sample particles, the stable starting position (x_as_) is one of the important indexes for the estimation. The x_as_ at the start of acceleration were around −50 μm for 10/5, −25 μm for 10/1, and −25 μm for 10/0.5, respectively. In the case of 10/5, the nanochannel width was relatively large so that the acceleration by electric field induced by the high electrical resistance of the nanochannel had already started before the nanochannel. Importantly, the acceleration values became unstable at 10/1 around the inlet of tapered focusing channel regions (See the position −60 < x < −25 μm in Figure 4d), probably because the influence of EOF remained in this region. On the other hand, such a flow instability was not observed at the case of 10/0.5. It suggests that the higher reduction ratio of tapered focusing channels, such as 10/0.5, is desirable for high acceleration, resulting in throughput improvement. 

In order to apply stronger electrophoresis for nanochannel devices, electrical blockage of small gas bubbles, which is generated by electrical reaction in the aqueous solution, should be controlled inside the nanochannel. As shown in the region of 0.8 V/−0.8 V and above DC bias voltage application, these phenomena are due to the instability caused by electrochemical decomposition of water inside the nanochannel. Similar issues have been reported in other analytical tools such as nanopores electrical detection and high performance liquid chromatography (HPLC). In order to address the issues, the electrophoresis electrode should be placed away from the nanochannel regions and/or that the degassed function parts should be integrated in the devices.

The stronger electrophoresis is not always desirable from the view of the signal detection and analysis. The stronger the electrophoresis is, the faster the sample molecules are passing so that high-speed translocation of sample behaviors reduces the obtained number of data-points. Therefore, the reduction of data-points makes identification difficult in analysis. Similarly, in the study of bio-nanopores that use electrophoresis for a flow control of sample DNA, the passing speed of single-stranded DNA is found to be several tens of microseconds per one base [6]. Since the sampling rate of the electrical measurement by combination of the cutting-edged A/D (Analog-to-Digital) converter and current amplifier is around 200 kHz, at least one millisecond per base is need because the number of data-points of one base affects the discrimination accuracy. The effect of electrophoresis on this device accelerates the transport rate of nucleobase molecules, as shown in Figure 4b, and therefore, the shape of nanochannel devices should be optimized for appropriate sample translocation speed. In this study, the average velocity was 1.22 milliseconds/base so that it meets this criterion.

## 5. Conclusions

We utilized electrophoresis to control the fluidity of sample biomolecules by using nanochannel-integrated nanogap electrode, and the throughput of single-molecule electrical signal detection was improved. EOF and thermal convection inside the nanochannel were suppressed by the addition of optimal concentration of the TBE buffer solution and the 0.1 *w/v*% surfactants of PVP. A tapered focusing channel region was inserted between the chamber and the nanochannel in the device, and the reduction ratio of inlet/outlet was optimized. We evaluated the velocity, acceleration, and capture region, and found that 10/0.5 has high electrophoresis performance in single-molecule measurement. Single-molecule measurement of dGMP was performed using this nanochannel-integrated device. It was found that the throughput was improved by nearly an order of magnitude compared to the case without electrophoresis. It was also found to reduce long-duration signals that could interfere with discrimination. This is because the strong electrophoresis flow inside the nanochannels prevents the molecules’ adsorption near the electrodes. This single-molecule electrical measurement with nanochannel-integrated nano-gap electrodes by electrophoresis significantly improved throughput of signal detections and identification accuracy. The silicon-based thin devices are flexible and the essential parts for the sensor are around 100 μm^2^ and are easily parallelized so that this single-molecule electrical detection method would be applicable for medical use, such as wearable sensors for medication [37,38] and for high density storage such as DNA digital data storage [39,40].

## Figures and Tables

**Figure 1 micromachines-11-00982-f001:**
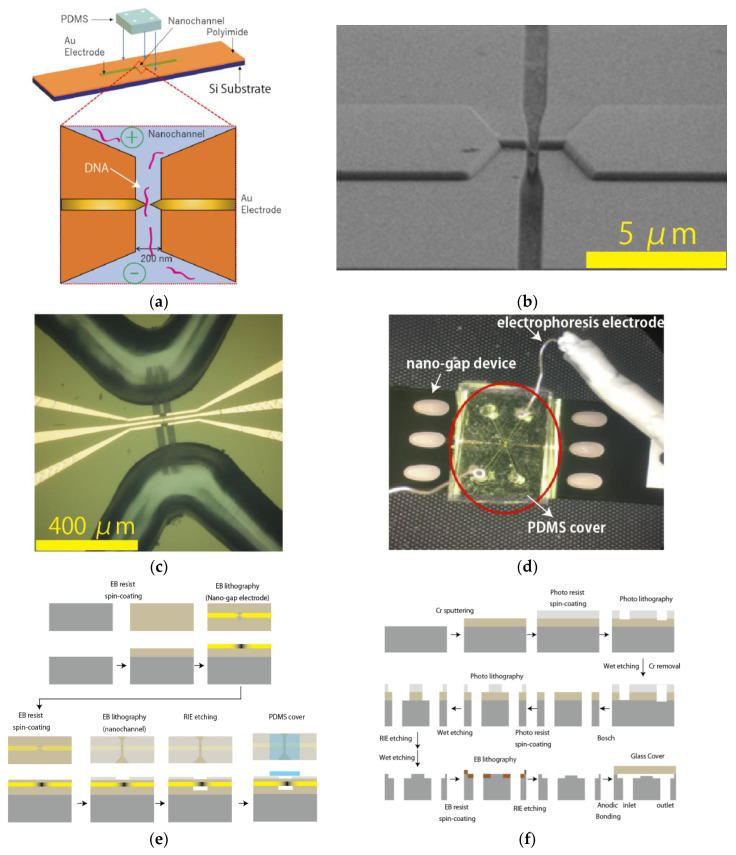
(**a**) Schematics diagram: A nanochannel-integrated nano-gap device that can control the fluidity of biomolecules such as DNA by electrophoresis. When DC bias voltage is applied across the solution chamber in the device, negatively charged DNA are translocating through the nanochannel and then nano-gap regions. The tunnel-current is induced via each of the sample molecules around the nano-gap electrode and so DNA molecules are detected as current positive signals. (**b**) Scanning electron microscope (SEM) image of a nanochannel-integrated nano-gap device, which has a nano-gap electrode (center), a nanochannel near the nano-gap, and a tapered focusing channel that connects microchannel/chamber regions and the nanochannel. The designed nanochannel has 200 nm width and 8 μm length, and the tapered focusing nanochannel has 2 μm inlet/0.2 μm outlet and 5 μm length. The accelerated voltage of electron is 2.0 kV and the 45-degree angled SEM. The yellow bar size is 5 μm. (**c**) Optical image (1000 × 1000 μm) of PDMS cover-fused nanochannel-integrated nano-gap device. The microchannel of the PDMS cover is connected to the nanochamber regions of silicon substrate, which have a squired-pillar region in the chamber in order to prevent a roof collapse of PDMS. The yellow bar size is 400 μm. (**d**) Electrophoresis set-up by a nanochannel-integrated nano-gap device. The white bar represents the scale of 10 mm. The PDMS have two microchannels (inlet/outlet) and sample solution chambers. In order to apply DC bias voltage between these two chambers, a pair of electrophoresis Ag/AgCl electrodes, which are made by an electrochemical oxidation of coiled silver wire, are inserted into the sample inlet/outlet hole of the PDMS. (**e**) Flow chart of Single-Molecule Electrical Device Fabrication Process. (**f**) Flow chart of Single-Molecule Imaging Device Fabrication Process.

**Figure 2 micromachines-11-00982-f002:**
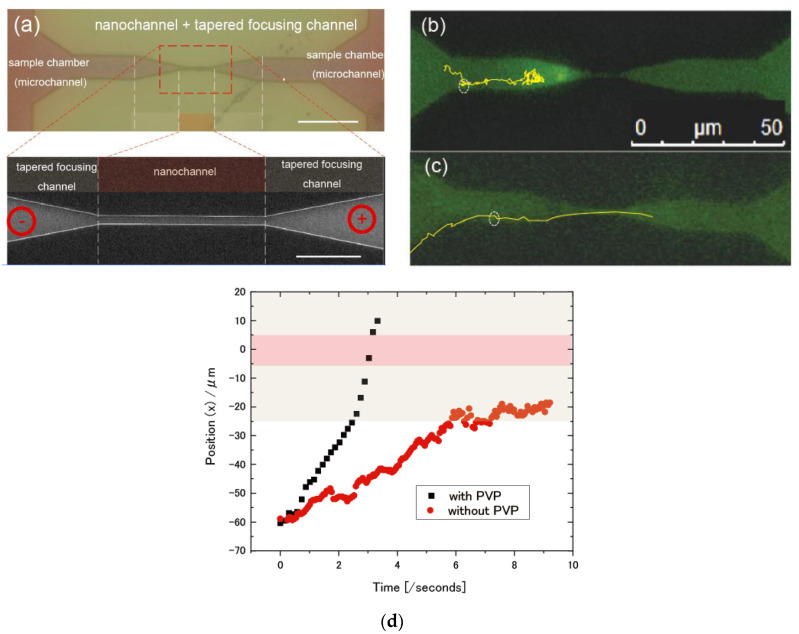
(**a**) An optical image of a nanochannel device for electrophoresis particle imaging. The lower image is the enlarged image of the red dot-line square area in the upper image. The scale bar for the upper image is 20 μm and that for the lower image is 5 μm. Three types of the 10 μm length nanochannel devices containing a 20 μm length of tapered focusing channel are fabricated. The reduction ratio of inlet width (μm) to outlet width (μm) are 10 μm/1 μm (10/1). The photo is the nanochannel containing a tapered focusing channel of 10/5. (**b**) We performed direct imaging of λDNA in nanochannel regions by optical microscope. The λDNA stained by YOYO-1, which is {1,1′-(4,4,8,8-tetramethyl-4,8-diazaundecamethylene)bis[4-[(3-methylbenzo-1,3-oxazol-2-yl)methylidene]-l,4-dihydroquinolinium] tetraiodide} and a green fluorescent dye used in DNA staining, is directly observed by fluorescent microscopy. Negatively charged DNA are translocating through the nanochannel and the tapered nanochannel regions. A typical image of YOYO-1-stained DNA in 0.05 × TBE solution containing 0.1 *w/v*% polyvinyl-pyrrolidone (PVP) under electrophoresis (bias DC voltage +5.0 V). The concentration of λDNA was 0.1 μg/mL. λDNA was easily flowing into nanochannel regions under this electrophoresis condition. (**c**) A typical image of YOYO-1-stained λDNA in 0.05 × TBE without PVP under electrophoresis of bias voltage +5.0. The behavior of sample λDNA was unstable inside tapered focusing nanochannel parts and backflow occurs especially around the inlet of the nanochannel. (**d**) A typical x position behavior of λDNA molecule in 0.05 × TBE solution containing 0.1 *w/v*% polyvinyl-pyrrolidone (PVP) (black) and without PVP (red), under electrophoresis (bias DC voltage +5.0 V). The λDNA stained by YOYO-1 is directly observed by fluorescent microscopy. The original movies for black and red dots are shown in Appendix A, respectively. λDNA was easily flowing into nanochannel regions under this electrophoresis condition for that with PVP (black dots). The sample λDNA was unstable inside tapered focusing channel parts and backflow occurs for that without PVP.

**Figure 3 micromachines-11-00982-f003:**
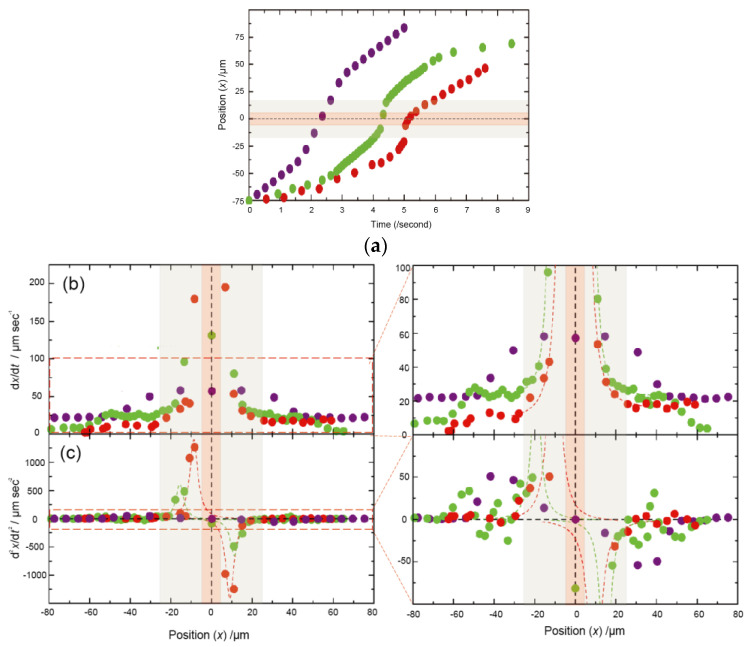
(**a**) Plot of the position (x) of sample particles for the time. We applied DC bias voltage of 5 V across the nanochannel devices. Negatively charged particles are translocating through the nanochannel device, and their positions (x) are plotted. The zero value of the position (x) represents the center of the nanochannel. The reduction ratios of inlet width (μm) to outlet width (μm) are 10 μm/5 μm (10/5: purple), 10 μm/1 μm (10/1: green), and 10 μm/0.5 μm (10/0.5: red), respectively. The orange area and the grey area in the graph show the nanochannel regions and the tapered focusing channel regions, respectively. Each of the dashed line represents the eye guideline for each of the *x* position traces. (**b**) The speed of particles is plotted for each position (x) in the upper left graph, and the acceleration of particles is plotted for each position (x) in the lower left graph. The speed of a particle is defined as the first derivative of particle position (dx/dt). The right upper-figure shows the enlarged graph of the orange dot-squared area of the left graph. (**c**) The acceleration of a particle is defined as the second derivative of particle position (d^2^x/d^2^t). Each of the maximum acceleration values were found to be 52 μm/s^2^ for 10/5, 486 μm/s^2^ for 10/1, and 1264 μm/s^2^ for 10/0.5, respectively. The right lower-figure is the enlarged graph of the orange dot-squared area of the left graph.

**Figure 4 micromachines-11-00982-f004:**
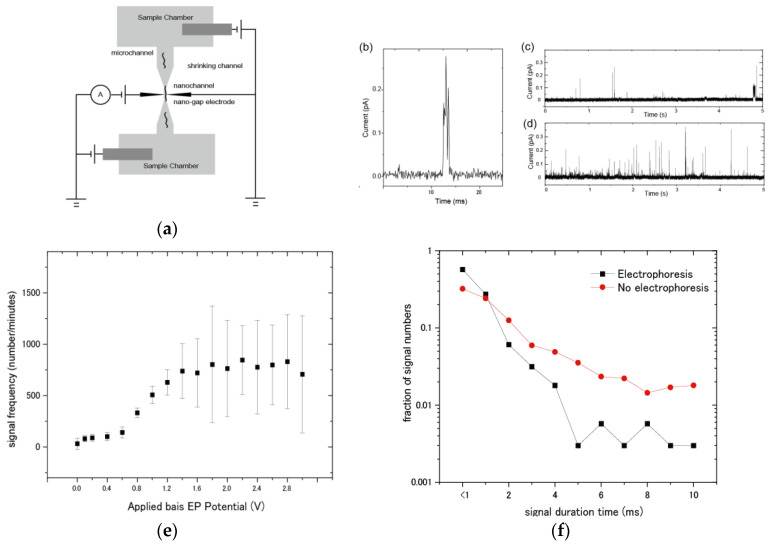
(**a**) Electrical diagram of single-molecule detection by electrophoresis with a nanochannel-integrated nano-gap device that can control the fluidity of biomolecules by electrophoresis. We applied electrophoresis between a sample chamber and another chambers as follows: The sensing electrode that is one of the gap electrodes was electrically grounded. Then, DC voltage was applied between a grounded sensing electrode (+0 V) and electrophoresis electrodes in the sample chamber and the same DC voltage was simultaneously applied between the grounded sensing electrode and electrophoresis electrodes in the other chamber. For instance, when −0.1 V for sample chamber and 0.1 V for another chamber were applied, the electrophoresis conditions are described as “0.2 V” because the potential difference between the chambers is 0.2 V. Tunnel current measurements were performed in 1 µM deionized aqueous solutions of sample deoxyguanosine monophosphate (dGMP) 0.05 × TBE containing 0.001% polyvinyl-pyrrolidone (PVP). (**b**) A typical current-time profile of dGMP signal. When sample dGMP molecules pass between the nano-gap electrodes, a tunnel phenomenon induces electron transfer through the molecules, resulting in the detection of individual electrical conductivity due to the electronic state of dGMP. (**c**) A typical I-t profile of dGMP signal before electrophoresis (no electrophoresis condition). (**d**) A typical I-t profile of dGMP signal after electrophoresis of 1.0 V. The electrophoresis condition of 1.0 V represents the DC application of +0.5 V between a grounded sensing electrode (+0 V) and electrophoresis electrodes in the sample chamber, and the simultaneous DC voltage application of −0.5 V between the grounded sensing electrode and another electrophoresis electrode in the other chamber, so that the bias potential became 1.0 V. (**e**) Dependency of electrophoresis with DC applied potential on signal frequency. Each of the present frequency values are defined as the signal number per ten seconds. The DC bias potential voltage for the electrophoresis conditions are 0, 0.1, 0.2, 0,4, 0,6, 0,8, 1.0, 1.2, 1.4, 1.6, 1.8, 2.0, 2.2, 2.4, 2.6, 2.8, and 3.0. Each of the total experimental times is six-hundredths of seconds for each of the applied bias potential conditions. Each of the error bars is defined as a standard deviation of signal numbers in each of ten seconds for at least sixty signal time-regions. (**f**) The fraction of signal numbers’ distribution for each of the signal duration times. The red and black dots represent the fraction of signal number under no electrophoresis, and that under electrophoresis of 1.0 V DC bias potential.

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
