# Peer review of "Single-Molecule Counting of Nucleotide by Electrophoresis with Nanochannel-Integrated Nano-Gap Devices"

_micromachines, 2020, doi:10.3390/mi11110982_

Round 1

Reviewer 1 Report

Summary of the manuscript

The manuscript by Ohshiro et al. reports the fabrication of a nanochannel structures for single molecule sensing through measurement of the tunelling current. Their geometry improves throughput through a reduction of electroosmotic flows within the channel via alteration of the electrolyte composition (reduction in the level of TBE buffer) and through the use of surfactants 0.001% polyvinylpyrrolidone.   They then further optimise the process through the use of a ‘shrinking channel’ geometry that allows for an electrophoretic concentration of the analyte molecules, analogous to the structures they have recently reported.

In my opinion, the manuscript represents a technical advance in the improvement of nanopore sensing, but there are some claims for which no data is provided, and other claims for which additional data is needed. I believe the work requires the following changes to be suitable for publication:

Major changes

  1. What is the size of the nanochannel? It should be explicitly mentioned in the Materials and Methods and the text for Figure 1.
  2. Modifications to the electrolyte buffer concentrations feature prominently in this paper, but the exact composition of the solution is never made clear. The authors should clearly state for each experiment (l DNA, beads, dGMP): the ionic salt concentration, the buffer concentration, and the surfactant concentration.
  3. Reducing the buffer concentration to 0.05 x TBE may affect the pH stability of the electrolyte solution. Could the authors provide data/references to show  that the pH buffering capability is unaffected?
  4. The addition of surfactant does appear to reduce the electroosmotic flow but may also affect the stability of the analyte, particularly proteins. The authors should discuss this point in the manuscript.
  5. The term ‘shrinking channel’ does not convey the idea the authors want to put across. I would suggest terming it a ‘tapered focussing channel’.
  6. Figure 1 – The schematic in (A) is unclear, what are the ‘blocks’ in the background? The schematic is also not drawn to scale, so a mention of scales could be avoided. It would be better to show a side-on view to allow the reader to understand how the sensing and analyte translocation occurs. This is already shown in Fig. 2(A), which makes this panel somewhat redundant. Sub-panel B also does not add any new information and can be removed. The scale bar in (D) is too small to read, please enlarge. The text for (D) appears after (E), please correct this. There are also typos in the legend.
  7. Figure 2 – this figure is very weak and needs additional data. The image in (B) is dark and does not add any information. This should be removed. The only evidence for reduction of electroosmotic flow is shown in panels (C) and (D), but this is indirect and there is no quantification. First, the authors should provide supplementary videos of the fluorescence imaging of DNA translocation to demonstrate clearly that there is reduced translocation, and also to make a claim for backflows. They must then show a line plot along the longitudinal axis of the device to highlight the difference in concentration of DNA on either side. Lastly, they must track the movement of DNA molecules through the channel as has been already done (Thacker et al., APL 2012 doi: 1063/1.4768929) to measure the effective electrophoretic constant, and show that this is substantially higher (due to the reduction in EOF) upon the surfactant treatment. This will be analogous to the data shown in Figure 3B. Other groups have also shown that at very low salt concentrations, such as that used by the authors, reversal of the electroosmotic flow is observed (Laohakunakorn et al. Nano Letters 2015 doi: 10.1021/nl504237k). The authors can use fluorescent carboxylate beads to verify if this occurs in their system. There are numerous typos in the legend and the figure.
  8. Figure 3 – sub-panel (A) should be added to Figure 2 instead. The authors must make clear how many particles were analysed for each condition in (B-D) and error bars should be provided for each datapoint. The concentration of the particles used is not mentioned in the text. What do the dashed lines represent and how were they calculated? Could the authors also state clearly which ratio was finally preferred in the main text at Line 210?
  9. Figure 4 – I presume the n=10 refers to the number of voltage combinations tested and not the number of points acquired to calculate the error bars. The latter number should be provided. Error bars should also be provided for sub-panel (B). The authors should provide a histogram of detection events to show that the changes in buffer concentration is not causing aggregation of the analyte.

Minor changes

  1. The nomenclature of (0.5V/-0.5V) used by the authors is cumbersome. Is there a reason for not quoting the potential difference instead, especially since the authors are only reporting symmetric combinations of positive and negative potentials?
  2. Lines 124 : the objective and its characteristics and camera used for imaging should be stated.
  3. Lines 141-145: unnecessary, and should be deleted.
  4. Lines 164-168: The sentences on Joule heating have no references and no data to support them. Indeed, many nanopore experiments pass small analytes through smaller pores under conditions of high ionic concentrations without reporting thermal decomposition. The authors must either remove these sentences and claims or provide evidence to back this up.
  5. Line 207: the first ratio mentioned should be ‘10/0.1‘.

Author Response

We would like to express many thanks to the referees for his or her valuable and helpful comments. We have studied the comments carefully and made the following revisions (The PDF file are uploaded in the box below).

Reviewer 2 Report

In this manuscript, the authors combined electrophoresis with a nanochannel 
integrated nanogap electrode 
to significantly improved throughput 
of single-molecule detection. I suggest accepting this manuscript after some minor edits.

  1. Add scale bars to all relevant figures.
  2. Add a cross sectional image of the device.
  3. Add a fabrication process flow chart for the fabrication section.

Author Response

We would like to express many thanks to the referees for his or her valuable and helpful comments. We have studied the comments carefully and made the following revisions. The PDF file are uploaded in the box.

Reviewer 3 Report

  1. Marking the different parts in figure 1 d and e will be more understanding for the readers
  2. What is the observation of the pressure required to push the solution through these nano channels? As the channels are very small it may need really high-pressure source to push the liquid through these channels- however if the pressure required for the liquid to push through nanochannel is high- then what is its effect on the cells/DNA mechanical stability?
  1. Single-molecule sensors with low voltage will also be interesting for wearable sensor application, in particular, application on pediatric for continuous monitoring it will be interesting to discuss this point by citing appropriate literature such as: "Liliana et.al- A Perspective on Microneedle-Based Drug Delivery and Diagnostics in Paediatrics" and "M Azmana et.al - Transdermal drug delivery system through polymeric microneedle: A recent update. Etc. 

Author Response

We would like to express many thanks to the referees for his or her valuable and helpful comments. We have studied the comments carefully and made the following revisions. The detail reply are shown in the box.

Round 2

Reviewer 1 Report

Replace 'trance' with 'trace at line 222 and line 231

Figure 1d: is the 'white bar' missing from the figure?

Line 366: 'is' not 'are'

Author Response

We would like to express many thanks to the referees for his or her comments. We have made the following revisions.
